# Neurophysiological Characterization of Thalamic Nuclei in Epileptic Anaesthetized Patients

**DOI:** 10.3390/brainsci9110312

**Published:** 2019-11-07

**Authors:** Lorena Vega-Zelaya, Cristina V. Torres, Marta Navas, Jesús Pastor

**Affiliations:** 1Department of Clinical Neurophysiology, University Hospital of La Princesa, Diego de León Street 62, 28006 Madrid, Spain; lorenacarolina.vega@salud.madrid.org; 2Department of Neurosurgery, University Hospital of La Princesa, Diego de León Street 62, 28006 Madrid, Spain; cristinatorresdiaz@yahoo.es (C.V.T.); martasoti@yahoo.es (M.N.)

**Keywords:** deep brain stimulation, epilepsy, extracellular action potentials, microelectrode recording

## Abstract

Deep brain stimulation (DBS) requires precise localization, which is especially difficult at the thalamus, and even more difficult in anesthetized patients. We aimed to characterize the neurophysiological properties of the ventral intermediate (V.im), ventral caudal (V.c), and centromedian parvo (Ce.pc) and the magnocellular (Ce.mc) thalamic nuclei. We obtained microelectrode recordings from five patients with refractory epilepsy under general anesthesia. Somatosensory evoked potentials recorded by microelectrodes were used to identify the V.c nucleus. Trajectories were reconstructed off-line to identify the nucleus recorded, and the amplitude of the action potential (AP) and the tonic (i.e., mean frequency, density, probability of interspike interval) and phasic (i.e., burst index, pause index, and pause ratio) properties of the pattern discharges were analyzed. The Mahalanobis metric was used to evaluate the similarity of the patterns. The mean AP amplitude was higher for the V.im nucleus (172.7 ± 7.6 µV) than for the other nuclei, and the mean frequency was lower for the Ce.pc nucleus (7.2 ± 0.8 Hz) and higher for the V.c nucleus (11.9 ± 0.8 Hz) than for the other nuclei. The phasic properties showed a bursting pattern for the V.c nucleus and a tonic pattern for the centromedian and V.im nuclei. The Mahalanobis distance was the shortest for the V.im/V.c and Ce.mp/Ce.pc pairs. Therefore, the different properties of the thalamic nuclei, even for patients under general anesthesia, can be used to positively define the recorded structure, improving the exactness of electrode placement in DBS.

## 1. Introduction

The human thalamus is a complex egg-shaped structure composed of more than 50 different nuclear groups [1]. Some of these groups, such as the anterior (Ant) or centromedian (Ce) groups, have been used to treat epilepsy by means of deep brain stimulation (DBS) [2,3,4,5,6,7]. The Ant nucleus is anatomically well isolated and is usually addressed by neuronavigation and imaging techniques, without the help of microelectrode recording (MER) or other neurophysiological techniques [8,9]. However, the Ce is surrounded by different nuclei, including the ventral intermediate thalamic nucleus (V.im), ventral caudal thalamic nucleus (V.c.), or parafascicularis (Pf).

The first requirement for achieving a good outcome after DBS, optimizing the power provided by the batteries and decreasing secondary effects, is to ensure the appropriate nucleus is addressed. The human thalamus, except for some nuclei (e.g., the previously mentioned Ant nucleus or geniculate nucleus), has no anatomical landmarks that permit the selection of a definite nucleus within millimeters. Therefore, it is extremely important to have neurophysiological clues that allow us to identify during MER, in a positive way, the nucleus being recorded. These clues are obviously more relevant in anaesthetized patients, in which communication with the patient is not possible. In addition to MER, other physiological tests can be performed during DBS surgery to identify nuclei, such as the cellular response to voluntary or passive movements of members, tactile stimulus, or paraesthesia induced by electrical stimulation [10,11]. However, most of these responses (except response to tactile stimuli) are unspecific, and all of them need conscious collaboration with the patient, and therefore cannot be assessed in anaesthetized patients. 

Recently, we showed [12] that the V.c. nucleus can be positively identified in anaesthetized patients by somatosensory evoked potentials (SSEP). Then, the nearby nuclei, namely, the V.im or Ce nucleus, can be identified during MER. Therefore, we can analyze neurophysiological properties and identify them even in anaesthetized patients.

For the reasons indicated above, DBS surgery is performed in awake patients to receive help from the patient during the activation maneuvers. This procedure is a fatiguing and stressful situation for the patient and for the surgical team, who are compelled to perform the surgery as fast as possible to avoid distress in the patient. Obviously, it would be preferable to do DBS in anaesthetized if we can ensure at least the same level of accuracy of anatomo-physiological localization in both anaesthetized and non-anaesthetized patients [13]. This method should be sufficiently robust to conserve some of the relevant physiological properties observed in awake patients during anesthesia. In other words, it must be invariant with respect to anesthesia.

The discharge properties of neurons depend on the set of ionic conductances (membrane excitability) and synaptic connections to other neural structures [14,15]. Both kinds of properties (excitability and connectivity) are highly specific for every neural structure. Therefore, we postulate that, although modified by anesthesia [16,17,18], some properties of connectivity and, in the strongest way, the properties of membrane excitability will be conserved and sufficiently different between several thalamic nuclei, allowing a positive identification of each nucleus.

The aim of our study was to characterize the neurophysiological properties of several thalamic nuclei, namely, the mean amplitude of the action potential (AP), mean frequency of the discharge, density of the neurons, and tonic and phasic properties of the discharge. Some authors have noted differences in the outcome between DBS stimulation of the centromedian magnocellular (Ce.mc) and centromedian parvocellular (Ce.pc.) nuclei [2,3]; therefore, we have analyzed both subdomains separately.

The preliminary results were published in an abstract form [19].

## 2. Materials and Methods

For this work, we followed the Schaltenbrand–Wahren (SW) atlas nomenclature [20]. Then, the structure referred to as the parafascicullaris/centromedian Pf/CM in the literature [21,22] is referred to as the Ce, and the ventro-posterolateral and ventro-posteromedial (VPL/VML) nuclei are referred to as the V.c. [23,24].

### 2.1. Patients and Surgery

We studied 5 patients (3 men, 2 women, Table 1) diagnosed with generalized epilepsy undergoing surgery for chronic DBS treatment in the Ce. All patients freely provided informed consent to participate in the procedures approved by the Hospital La Princesa Ethics Board.

The patients were initially assessed for study eligibility using a presurgical evaluation in our center [25,26], and patients undergoing resective surgery were excluded. Most of the patients were treated using vagus nerve stimulation (VNS), but after a two-year period of inadequate results, DBS in the Ce nucleus was proposed. During surgery, VNS was turned off and, immediately after recovery, the stimulations were continued at the same schedule.

All patients were operated on while under general anesthesia using propofol (5.48 ± 0.28 mg/kg/h, [4.5,6.2]) and remifentanil (0.12 ± 0.02 µg/kg/min, [0.1,0.2]), maintaining a bispectral index (BIS) between 40 and 45. Neuromuscular blocks were performed with cis-atracurium (0.5 mg/kg). No changes in anti-epileptic treatment was performed during the surgery.

The thalamus was identified using a 1.5 T magnetic resonance imaging (MRI) scanner (General Electric®, Fairfield, CT, USA), and the coordinates were located stereotactically with a neuronavigator (BrainLab®, Feldkirchen, Germany). The coordinates were calculated by fusing the MRI images and CT scans, according to the SW atlas. For thalamic DBS electrode placement, a tentative initial target was selected in the Ce (x = 8, y = −10, z = 0). All the coordinates corresponded to the mid-intercommissural AC-PC line (anterior commissure-posterior commissure). Neuronal recordings (Leadpoint®, Minneapolis, USA) were obtained, starting at 10 mm above the target and progressing in increments of 0.5 mm. Microelectrode recordings (MER- FHC®, Maine, USA) were obtained for both spontaneous and sensory-evoked activity until the inferior border of the thalamus was confirmed by the absence of neuronal activity. The impedance was always higher than 900 kΩ (1696 ± 80 kΩ, (900,2900)).

MERs were obtained through four microelectrodes placed 2 mm apart and placed (usually) at anterior, center, posterior, and lateral locations; in patient 5 only, the posterior electrodes were replaced by medial electrodes. Neuronal recordings (Leadpoint®, Minneapolis, USA) of 15–90 s duration were obtained, beginning 10 mm above the target and progressing in steps of 0.5 mm. MER were obtained from both spontaneous and sensory-evoked activity until the inferior border of the thalamus was confirmed by the absence of neuronal activity.

The Microdrive was fixed to a stereotactic Leksell Coordinate Frame (Elekta®, Stockholm, Sweden). The bandwidth for spontaneous activity was 200 Hz–5 kHz, and the sample rate was 24 kHz; for the SSEPs only, the bandwidth was 2 Hz–5 kHz. A notch filter was not used. However, in two trajectories, the recordings were performed with the notch filter (50 Hz).

After the Ce was identified, a quadripolar DBS electrode was implanted. During the same surgery, the leads were connected to an implanted programmable stimulator (Kinetra, Medtronic® in three cases and Libra XP, Saint Jude® in two) placed in a pectoral or abdominal location.

### 2.2. Somatosensory Evoked Potentials

Somatosensory evoked potentials started at 8 mm above the theoretical target and repeated at 1 mm intervals until the end of the thalamus was reached. The lower border of the thalamus was easily identified by the absence of AP. Two rounds of 500 pulses were delivered at each point. The stimulation was elicited by electrical stimulation of the contralateral median nerve at the wrist at an intensity 1.5 times higher than the motor threshold (defined at the beginning of the surgery) with 200 μs pulses at a frequency of 7.1 Hz [19]. Responses at the cervical point (C2-Fpz) and both parietal regions (C3’/C4’-Fpz) were recorded by 18 mm subdermal needles (SGM®, Ljubiceva, Croatia) with a bandwidth of 10–1500 Hz (notch filter off). Simultaneously, the activity recorded through the four microelectrodes was averaged across the trials (Appendix A, Figure A1a).

The presence of high frequency oscillations (HFO) of approximately 800 Hz was used as a specific landmark of the presence of the V.c nucleus (Appendix A, Figure A1a). A detailed analysis of these responses is beyond the scope of this manuscript (see [12]).

### 2.3. Reconstruction of the Trajectory

A detailed description of the reconstruction can be found in a previous study [12]. In brief, we consider coordinate *z* = 0 the last recording inside the thalamus defined by the presence of an AP. The anteroposterior and lateral coordinates were obtained from the postoperative MRI scans performed one month after the surgery. Using this point and the stereo-tactic angles, we reconstructed the real trajectory of the electrode in a three-dimensional space in 1 mm intervals. Therefore, using the SW map, we were able to identify in which nucleus each electrode was located throughout a trajectory (Appendix A, Figure A2).

### 2.4. Analysis of Discharge Properties

We collectively analyzed the properties of discharge. To identify the neurons, we performed a sorting spike that, briefly, included the next steps. Data were exported to ASCII files, and analyses were performed off-line with bandwidth 0.5–5 kHz. Action potentials were identified and grouped according to these properties: Maximum and minimum amplitude, maximum and minimum value of the first derivative and duration of depolarizing and repolarizing phases. Using Mahalanobis metric, we clustered AP from the same neuron [12]. We can distinguish between tonic and phasic properties [24]. The following tonic properties were identified:Amplitude of the AP, measured from peak to peak (in µV). This property is not a tonic property, but is commonly used in clinical practice; therefore, we included it in this group.Mean frequency and standard deviation of the raw trace and for every neuron. Both values were obtained from the inverse of the instant frequency.Density, defined as the number of cells recorded by every electrode at one position. AP sorting was performed by clustering by using the Mahalanobis distances (see below) of several properties of the AP (e.g., amplitude and duration of positive and negative phases and maximum and minimum value of the first derivative). The maximum number of cells allowed to be identified by this mean was chosen as 5.Probability density functions (pdf) of the inter-spike interval (ISI) for every neuron. Relative frequency was computed for 1 ms bins, and the probability/bin (*p_i_*) was calculated with the following expression:(1)$$p_{i}=\frac{f_{i}}{\sum_{j=1}^{N}f_{j}}$$
where *f_i_* is the frequency for the *i-bin* and *N* is the total number of bins.

To study the structure of these inter-spike interval histograms, we fitted the data to a double exponential decay function, using the following expression:(2)$$y\left(x\right)=\alpha +\beta e^{-\gamma x}+\delta e^{-\varepsilon x}$$
where α,β,γ, δ, and ε are the constants to be fitted. The least square sum method was used to fit the empirical data to the defined functions.

The properties of phasic activity analyzed were as follows [27,28]:Burst index (BI) is defined as the ratio between the number of ISI < 10 ms and the number of ISIs > 10 ms. It represents the number of bursts of discharges with respect to individual discharges, and is calculated as follows:
(3)$$BI=\frac{N_{ISI<10ms}}{N_{ISI>10ms}}$$Pause index (PI), defined as the ratio between the number of ISIs > 50 ms and the number of ISIs < 50 ms, is calculated as follows:
(4)$$PI=\frac{N_{ISI>50ms}}{N_{ISI<50ms}}$$Pause ratio (PR), defined as the total duration of pauses (ISI > 50 ms) divided by the total duration of no-pauses (ISI < 50 ms). Although similar in name, the information obtained is different from that of the PI, and the PR is calculated as follows:
(5)$$PR=\frac{\sum_{i=1}^{N}ISI_{>50\;ms}^{i}}{\sum_{j=1}^{N}ISI_{<50\;ms}^{j}}$$

Considering that the mean frequencies obtained were lower than those in awake patients, we computed all of these measurements using different pairs of threshold times instead of 10/50 ms. In fact, we attempted to use 20/100 and 50/400 ms.

All analyses were performed in custom MATLAB®R2018 scripts.

### 2.5. Evaluation of Global Similarity

Global comparisons between different nuclei cannot be performed adequately with pairs of variables. We wanted to ascertain the distance (e.g., dissimilarity) between all four nuclei considering all the variables. We measured every nucleus as a point in a 6-dimensional space, defined as follows:
(6)nuc=(amplitude,frequency, density,BI, PI, PR);nuc={Ce.mc, Ce.pc,V.im,V.c}

Obviously, we cannot represent a 6-dimensional space in a two-dimensional figure. However, we can calculate the distance between every pair of nuclei. Therefore, the smaller the distance is between a pair, the more similarity there is between the nuclei. To that end, we used the Mahalanobis distance [29]. First, we computed the covariance matrix (S) with variances ($s_{i}^{2}$) on the main diagonal. For every pair of nuclei (*i,j*), we computed the distance ($d_{ij}$) according to the following expression:(7)$$d_{ij}={\left[\left(nuc_{i}-nuc_{j}\right)S^{-1}{\left(nuc_{i}-nuc_{j}\right)}^{\prime }\right]}^{1/2}$$
where ${\left(nuc_{i}-nuc_{j}\right)}^{\prime }$ is the transpose vector of $\left(nuc_{i}-nuc_{j}\right)$.

### 2.6. Statistics

Kurtosis (*K*) was computed for every group, and only signals with values between 2 and 8 were considered signals from only one group [30]. Extreme outliers were removed progressively until *K* < 8. Statistical analysis was applied only to these homogeneous groups. 

Statistical comparisons between groups were performed using the z-score, Student’s t-test, or ANOVA for data with normal distributions, and for data that failed the normality tests, the Mann–Whitney Rank sum test or Kruskal–Wallis analysis of variance were used. Normality was evaluated using the Kolmogorov–Smirnov test. SigmaStat® 3.5 software (Point Richmond, CA, USA) was used for statistical analysis. Instead of using rank as a dispersion measure, we used a 25–75 interquartile range, which is shown between brackets. In this context, a group refers to a set of measures of the same variable obtained from the same nucleus, e.g., mean amplitude AP, frequency, BI.

Pearson’s correlation coefficient was used to study the linear dependence between the variables. Linear regression significance was evaluated by a contrast hypothesis against the null hypothesis *ρ* = 0 using the formula:
(8)$$t=\frac{r\sqrt{N-2}}{\sqrt{1-r^{2}}}$$
which describes a Student distribution with *N* − 2 degrees of freedom.

The significance level was set at *p* = 0.05. The results are shown as the mean ± SEM, except where otherwise indicated. As stated above, the 25–75 interquartile range is reported between brackets, and the median is referred to as *Med*.

## 3. Results

The presence of high frequency oscillations (approximately 800 Hz) and the reconstructed trajectories allowed us to identify the nuclei where raw recordings were coming up (Appendix A, Figure A1 and Figure A2), assigning a cumulative trajectory (computed from all the tracks and from all the patients) of 171 mm to the V.c., 99 mm to the Ce, and 83 mm to the V.im. This method [12] has been proven to be robust in the identification of different nuclei and is extremely important for our purpose to avoid biases from the mixing of recordings from different nuclei.

The presence of a raw MER between 15 and 90 s at each point ensured that we had a significant number of extracellular APs. For every nucleus, we recorded a minimum of 1100 action potentials and a minimum of 41 traces. Examples of raw traces for the Ce, V.im, and V.c are shown in Appendix A, Figure A2.

### Tonic Properties

These properties are commonly reported in the literature and are essentially the only properties used in the clinical assessment of thalamic nuclei, even though other mathematical tools have been used in previous reports [27,28,31]. However, neither the complete list of nuclei addressed in this manuscript, nor the level of detail analyzed, has been previously described in humans. 

Usually, the amplitude of the AP is not considered for the clinical identification of nuclei in part because the variation in the amplitude is excessively high. However, we compared this property for the four thalamic nuclei addressed in this work. The mean AP amplitudes for Ce.mc, Ce.pc, and V.c were not different (82.7 ± 2.4, *Med* = 86.9, (75.4–95.2), 93.3 ± 3.6, *Med* = 91.3, (88.1–103.6), and 133.4 ± 3.8, *Med* = 136.9, (124.4–142.3) µV, respectively), but the mean AP amplitude was higher for V.im (*p* < 0.01) than for the other nuclei (Figure 1a), as it showed an amplitude nearly double that of the centromedian nuclei (172.7 ± 7.6, *Med* = 169.1, (156.6–191.6)).

Not only the AP amplitude was different for the different nuclei, but the firing rate was also different (Figure 1b). In fact, there were low values for both centromedian subdomains (7.2 ± 0.8, *Med* = 10.5, (5.2–14.8) and 9.4 ± 0.8, *Med* = 8.1, (5.0–11.4) Hz for the Ce.pc and Ce.mc, respectively), which were not different. The frequency was high for the V.im (10.3 ± 0.6, *Med* = 10.8, (8.7–12.9) Hz, significantly different from the Ce.pc –*p* < 0.05, but not from the Ce.mc) and was the highest for the V.c (11.9 ± 0.8, *Med* = 11.8, (9.8–13.7) Hz –*p* < 0.01), even though the frequency of the V.c was similar to that of the V.im and different from those of both centromedian nuclei (*p* < 0.001).

The density of the cells recorded in every nucleus (Figure 1c) was similar for the Ce.pc, V.im, and V.c (2.36 ± 0.08, *Med* = 2.1, (1.98–2.56), 2.27 ± 0.08, *Med* = 2.22, (2.19–2.42), and 2.29 ± 0.07, *Med* = 2.35, (2.25–2.42) cells/trace, respectively), but it was lower for the Ce.mc (2.10 ± 0.09, Med = 2.01, (1.89–2.34) cells/trace –*p* < 0.05).

The inter-spike intervals for the different nuclei were pooled and are shown in Figure 2. 

All the histograms were well fitted by double negative exponential functions, following Equation (2) (see Appendix B). It is important to observe that all the correlation coefficients are above 0.990. 

Finally, to characterize the phasic properties of the thalamic nuclei, we studied the BI, PI, and PR. All these properties are obtained from the raw pattern discharge, and are usually described by a binary plot (Figure 3) and inter-spike interval histograms. Periods between consecutive APs can be easily computed from binary plots, where each AP is represented by a vertical line.

We used three different pairs of threshold times to compute the phasic variables. Therefore, we speculated that other pairs of times, different from 10/50 ms, could yield better results, taking into account that the mean frequencies obtained were lower than those published for awake patients. However, Figure 4 shows that the results are different for the BI and PI/PR. In fact, the discriminant capacity for BI is clearly lower for the 10/50 ms pair than for the other pairs. However, for PI and PR, the most discriminant pair threshold is 10/50 ms. 

In addition, the behavior of the BI for different pairs of thresholds is worse than that of the PI. In fact, a clear linearity can be observed for the three pairs of thresholds in the PI, but not for those in the BI. For the PR, only the 10/50 ms threshold pair shows linearity (Appendix C). Therefore, we used the 20/100 ms threshold pair for the remainder of the work.

As shown in Figure 5, the BI for the V.im nucleus (6.479 ±1.165, *Med* = 4.911, (3.008–5.866)) is higher than those for the Ce.pc and Ce.mc (4.716 ± 2.103, *Med* = 2.465, (1.799–3.125) and 3.744 ± 0.317, *Med* = 3.749, (2.521–4.927), *p* < 0.05 and *p* < 0.01, respectively) but is not different from that of the V.c (5.633 ± 0.317, *Med* = 4.964, (3.533–7.096)). Nonetheless, the BI for the V.c is also higher than that of the Ce.pc (*p* < 0.05). The PI and PR were higher for the Ce.pc (0.278 ± 0.025, *Med* = 0.26, (0.169–0.368), 0.648 ± 0.065, *Med* = 0.586, [0.366–0.867] respectively for PI and PR) than for the V.im (0.167 ± 0.012, *Med* = 0.154, (0.120–0.226) and 0.367 ± 0.029, *Med* = 0.331, (0.253–0.501), *p* < 0.01) and the V.c (0.149 ± 0.068, *Med* = 0.140, (0.104–0.195), and 0.324 ± 0.016, *Med* = 0.298, (0.218–0.427), *p* < 0.001), but was not higher than that for the Ce.mc (0.219 ± 0.022, *Med* = 0.195, (0.153–0.258) and 0.496 ± 0.057, *Med* = 0.429, (0.327–0.581)).

Finally, we addressed the similarity/dissimilarity between the global properties of the nuclei by using the Mahalanobis distance (*d*) according to Equation (6). The sorted relations between pairs of nuclei are V.c–V.im (*d* = 2.178) < Ce.pc–Ce.mc (*d* = 3.086) < Ce.mc–V.c (*d* = 3.1857) < Ce.mc–V.im (*d* = 3.617) < Ce.pc–V.c (*d* = 4.038) < Ce.pc–V.im (*d* = 4.266). Therefore, the two nuclei that have a shorter distance (e.g., the two nuclei that are more similar) are V.im and V.c, followed by Ce.pc and Ce.mc, which are clearly separated from V.c and V.im.

## 4. Discussion

We have shown in this work that the global properties of different thalamic nuclei in anaesthetized patients are different and can be used to identify the nuclei. The centromedian nucleus can be easily identified during MERs and distinguished from the other thalamic nuclei. These facts are relevant from the clinical point of view because this set of properties can be used to identify every nucleus with precision by MERs during DBS surgery.

Currently, the mean frequency discharge and the expertise of the neurophysiologist in identifying a raw record as phasic or tonic are the main tools used during MERs to identify the structure recorded. In awake patients, some other clues can be used, such as the response to active or passive limb movements, the presence of paresthesia and a neural response to light tactile stimulation [10,11]. Obviously, all these physiological activations cannot be used in anaesthetized patients. Moreover, these responses are unspecific to a considerable degree (except for the neural response to light touch), and therefore, a positive identification of diverse nuclei in awake patients is of great importance.

From a methodological viewpoint, it is important to consider that all patients were under general anesthesia, and this condition can modify neuronal discharge features and perhaps the synaptic functions of some thalamic nuclei [32], but the exact degree of the effect remains to be elucidated. However, it is known that the mean frequency discharge (a useful variable during MERs) [16,17,18,33] and other electrophysiological properties, such as local field potentials [34], are affected by anesthesia. However, there is evidence in a study of DBS of the subthalamic nucleus (STN) that localization with MER and patient comfort can be achieved with equal effectiveness in patients under generalized anesthesia compared with that in awake patients [35].

Another important consideration is that we have used a reconstruction of the real trajectory [12] to identify the placement for every electrode that can be plotted clearly in the SW planes. This approach can only be performed off-line because we need to identify the lower border of the thalamus. However, this approach has been shown to be sufficiently robust to identify nuclei, enabling raw traces from different patients pertaining to the same neural structure to be grouped.

The AP amplitude was different only for the V.im nucleus; the rest of the nuclei were analyzed, but parts of centromedian and V.c could not be distinguished. Extracellular recordings retain some of the properties of APs acquired intracellularly [15,36,37], but amplitudes of extracellularly recorded APs do not depend solely on features of membrane excitability. In fact, the amplitudes of extracellular APs decrease rapidly as a function of the distance between the tip of the recording electrode and the soma [37]. However, the relative position between the electrode and soma is not the important factor. In addition, the local field potential depending on the contribution of the global sources of the environment can modify the amplitude [36]. The duration of an AP is a more robust measure that is not affected by any of these variables; therefore, we postulate that measurements of duration and other properties of an AP (number of phases, durations of them) can be more useful in identifying nuclei. This idea remains to be elucidated by future studies.

Obviously, the number of ISIs lower/greater than a defined value (e.g., 10, 50 ms) depends on the pdf, which is related (not in a linear way, of course) to the mean frequency. In awake patients, the mean frequency described is higher than that in the patients in our study; therefore, the number of ISIs < 10 ms should be greater than that in the patients in our study [17,18,28]. We have shown that for lower frequencies, such as those obtained in anaesthetized patients, the 20/100 ms pair of threshold times is preferable to the pair of lower threshold times, 10/50 ms, used in awake patients [28].

We have characterized every nucleus by a point in a 6th dimensional configuration space. This approach is typical of multivariate analysis in which every dimension represents different magnitudes and physical units. Therefore, every point describes the state of the system. It’s quite useful to assess how close or far away two systems are, i.e., two points of this space; in other words, we need to describe a metric [38]. The Mahalanobis distance is a useful multivariate analysis tool used to measure the distance between a pair of points in a multidimensional space. It is based on Minkowski’s distances, but in contrast to these, it does not depend on the existence of natural units of measure [39,40]. As we can observe from Equation (7), this metric is dimensionless and therefore, there are not inconsistencies in its use. We have shown that the similarity is high between the V.c and V.im, and is surprisingly low between both parts of the centromedian. The distance is large between the centromedian and V.im and even larger between the centromedian and V.c. The V.im nucleus has a high AP amplitude and a mean frequency between those of the centromedian and V.c, but some properties (e.g., PI and PR) are similar to those of the V.c.

Properties related to pattern discharge are scarcely used in clinical practice, but they have been analyzed for different structures addressed during DBS, such as the STN [41,42], Ant thalamic nuclei [43], and globus pallidus [21,44]. However, these properties have not been described for the nuclei analyzed in this work. We have shown here that even under general anesthesia, some properties of raw discharge are different for these thalamic nuclei. This is a very important fact, irrespective of whether these properties share a similar relationship in awake patients, because they can be used to provide a positive identification of thalamic structures only by MER without the need for conscious patient collaboration. However, other properties, as mean pdf (Figure 2), are not able to discriminate between nuclei. We cannot exclude a bias derived either from the small number of patients or an anesthetic effect. We need a bigger number of patients, and comparison with pdf obtained from awake patients to explain this fact. In a complex structure such as the thalamus, the placement of electrodes a couple of millimeters apart can involve a completely different nucleus. Hence, identifying the position at which the electrode is placed to the most accurate extent possible is extremely important. During DBS surgery, drainage of cerebro-spinal fluid or air entering inside the cranium can displace the target as much as several millimeters from its theoretical coordinates [45,46]. Identification of a neural structure by its electrophysiological properties prevents errors from occurring due to this morphological change of the brain.

In summary, it is important to understand that our data were obtained from a small number of patients. Although these data seem to be robust, we need a larger cohort to unequivocally establish these results.

## 5. Conclusions

In this work, we demonstrated that properties related to raw traces are different for different thalamic nuclei in anesthetized patients. Isolated single features (i.e., amplitude or frequency) cannot sufficiently discriminate between nuclei to identify a raw trace. However, it may be relatively easy to compute the properties described in this paper online to identify the structure from which the discharge is obtained. We have characterized the electrophysiological properties of the centromedian nucleus, which have not been previously described. We propose that this method can increase the accuracy of electrode placement in DBS and decrease the secondary effects and energy consumption derived from the incorrect placement of electrodes.

## Figures and Tables

**Figure 1 brainsci-09-00312-f001:**
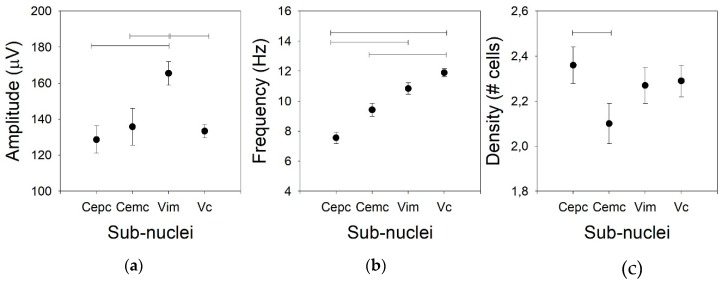
Tonic properties for the four nuclei. (**a**) Amplitude of the action potential (AP), (**b**) mean frequency of the raw traces, and (**c**) density of the cells. Horizontal lines indicate statistically significant differences between pairs of nuclei based on the Kruskal–Wallis analysis of variance test.

**Figure 2 brainsci-09-00312-f002:**
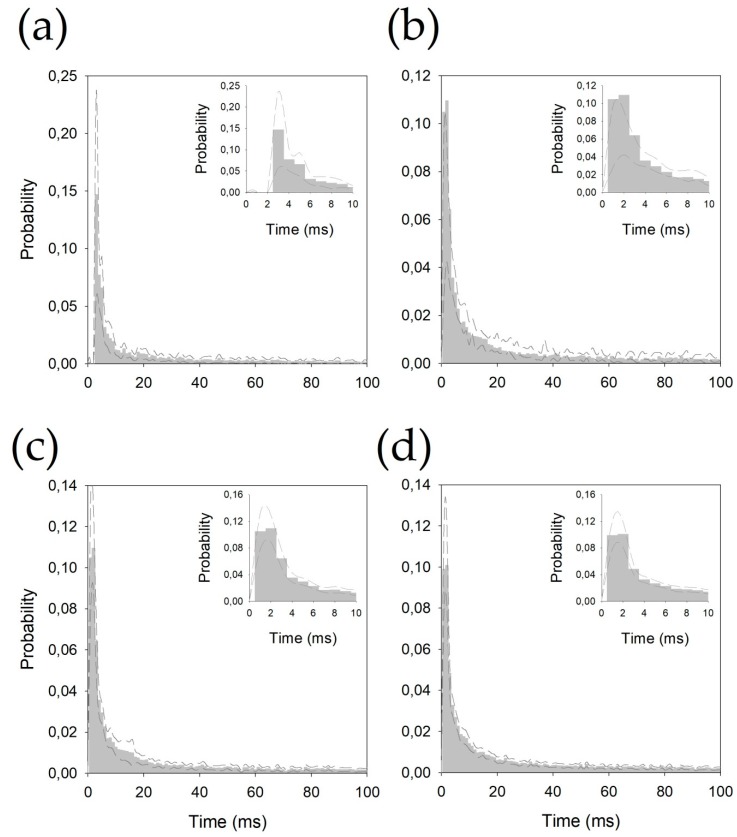
Probability density functions pooled for (**a**) centromedian magnocellular (Ce.mc), (**b**) centromedian parvocellular (Ce.pc) (**c**) ventral intermediate thalamic nucleus (V.im), and (**d**) ventral caudal thalamic nucleus (V.c). Discontinuous lines represent ± 2.5 SEM. Insets show the details of the first 10 ms.

**Figure 3 brainsci-09-00312-f003:**
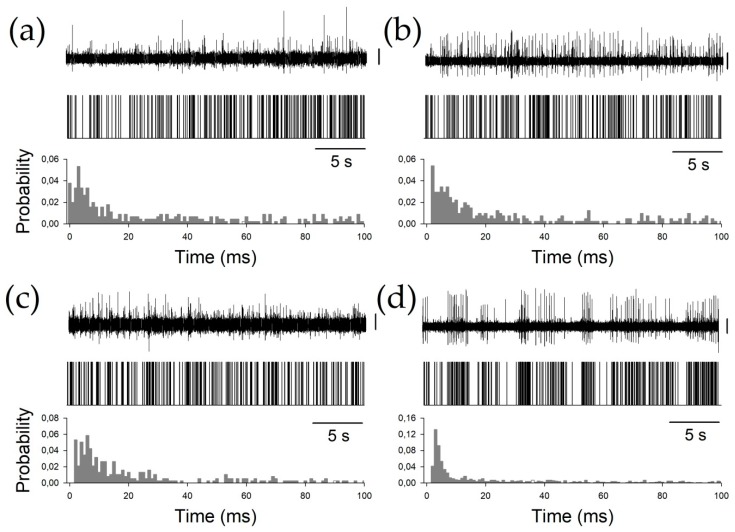
Examples of all the nuclei. (**a**) Ce.pc, (**b**) Ce.mc (**c**) V.c, and (**d**) V.im. For every panel, the upper row shows the raw trace obtained by the microelectrode recording (MER), the middle row shows the binary plots, and the bottom row shows the probability density functions (pdfs). Vertical calibration bars = 50 µV.

**Figure 4 brainsci-09-00312-f004:**
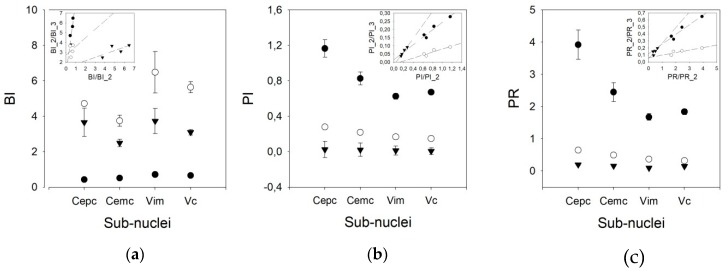
Graphs showing the values of (**a**) burst index (BI), (**b**) pause index (PI), and (**c**) pause ratio (PR) for all the nuclei for different pairs of times (solid dot = 10/50 ms, empty dot = 20/100 ms, and triangles = 50/200 ms). Insets represent linear regressions for the values obtained for all the nuclei using different pairs of times (solid dots = 10/50 vs. 20/100, empty dots = 10/50 vs. 50/200 and triangles = 20/100 vs. 50/200).

**Figure 5 brainsci-09-00312-f005:**
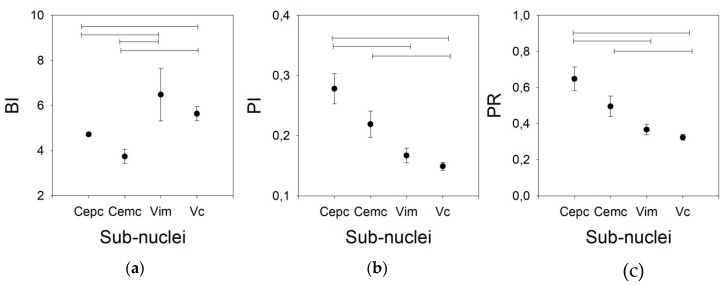
Phasic properties for the four nuclei for the 20/100 ms threshold pair. (**a**) BI, (**b**) PI, (**c**) PR. Horizontal lines indicate statistically significant differences between pairs of nuclei according to the Kruskal–Wallis analysis of variance test.

**Table 1 brainsci-09-00312-t001:** Clinical features of the group studied.

Pat.	Sex	Age (years)	History (years)	Etiology	v-EEG	MRI Result	VNS	AED
1	F	37	31	Genetic ^1^	GE	Normal	Yes	PGB, CBZ, CNZ
2	F	18	12	LGS	GE	Dysplasia LF	No	RUF, VPT
3	M	34	27	Genetic ^2^	EG/EE	Normal	Yes	VPT, PGB, LAC, ZNS
4	M	27	27	LGS	GE	Normal	No	LVZ, OXC, LAC, CZM
5	M	30	23	Structural	GE/EE	Dysplasia biFT	Yes	TPM, VPT

AED: Anti-epileptic drugs. biFT: Bilateral fronto temporal. CBZ: Carbamacepin. CNZ: Clonacepam. CZM: Clobazam. F: Female. M: Male. EE: Epileptic encephalopathy. GE: Generalized epilepsy. LAC: Lacosamide. LF: Left frontal. LGS: Lennox-Gastaut syndrome. LTG: Lamotrigine. LVZ: Levetiracetam. OXC: Oxcarbamacepin. PGB: Pregabalin. RUF: Rufinamide. TPM: Topiramate. VPT: Valproate. ZNS: Zonisamide. ^1^ 20 ring-chromosome syndrome. ^2^ Tuberous sclerosis.

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
