# Peer review of "Neurophysiological Characterization of Thalamic Nuclei in Epileptic Anaesthetized Patients"

_brainsci, 2019, doi:10.3390/brainsci9110312_

Round 1

Reviewer 1 Report

The paper “Neurophysiological characterization of thalamic nuclei in anaesthetized humans” provides a neurophysiological characterization of the thalamic nuclei using micro electrode recordings (MER) in anesthetized patients with epilepsy undergoing surgery for deep brain electrodes implantation for deep brain stimulation (DBS) surgery. Despite the interesting idea of characterizing the different nuclei to improve accuracy in target identification during DBS surgery in anesthetized patients, the paper requires several clarifications and improvements:

The title should clearly identify the patients involved in the study (epileptic patients). This is because there are other studies characterizing thalamic nuclei during DBS targeting in other pathologies in anesthetized patients (e.g. Marceglia et al Mov Disord 2010) The mean amplitude (mentioned since the introduction) is a questionable measure because it depends on the spatial relationship between the recording dipole (electrode) and the source dipole (neuron). The recording procedures should all be reported in the Methods section (there is the length of 15-90 s that is reported in the results). Also the descending step on the trajectory is not defined. The phasic activity properties definition come from an analysis on Parkinson’s patients who are characterized by bursting activity (regular and irregular) in the beta band. It is therefore correct to try other thresholds, but these thresholds should be defined according to the characteristics of the pdf which represent the characteristics of the activity. The global comparison between nuclei is quite unclear. Each nucleus is represented by a set of 6 variables, but these variables are different. The amplitude is a characteristic of the single neuron, the frequency is a characteristic of a neuron in a trace, the density is a characteristic of a trace, the mBI is not defined, the PI and PR are properties coming from the pdf of a single neuron, but could also come from the pdf of pooled neurons. Therefore, it is impossible to understand whether these are parameters or a set of random variables. From here also comes the definition of the covariance matrix. An exemplary representation of a nucleus should be provided. The statistical analysis talks about “groups” but this term is not referred to anywhere else. Please explain. Lines 195-200 report the details of the statistical tests performed across “groups”. However, no p-values are reported in the Results. The pooled pdfs seem quite similar, and do not discriminate between nuclei. As shown in Figure 3, the burst index is not informative because it is defined in such a way that requires the bursts to be present, but in many traces there are no bursts. Regular burts are represented by a pdf with a biphasic distribution (one for the rhythm of the burst and the other one for the rhythm of the spike). This may explain why BI has high variability in the results.

Minor points:

In the introduction, Ce.m.c. and Ce.p.c. are not defined (only Ce is defined) Line 248: interstimulus interval should be inter-spike interval Figure 3 refers to the raw trace without spike sorting, correct? Lines 262-264: revise the sentence. There is a missing verb.

Author Response

We acknowledge the reviewer by the incisive questions raised and the goal in clarifying the manuscript. He/she has done a clever and detailed analysis of the manuscript.

Comments and Suggestions for Authors

The paper “Neurophysiological characterization of thalamic nuclei in anaesthetized humans” provides a neurophysiological characterization of the thalamic nuclei using micro electrode recordings (MER) in anesthetized patients with epilepsy undergoing surgery for deep brain electrodes implantation for deep brain stimulation (DBS) surgery. Despite the interesting idea of characterizing the different nuclei to improve accuracy in target identification during DBS surgery in anesthetized patients, the paper requires several clarifications and improvements:

The title should clearly identify the patients involved in the study (epileptic patients). This is because there are other studies characterizing thalamic nuclei during DBS targeting in other pathologies in anesthetized patients (e.g. Marceglia et al Mov Disord 2010). We have changed the title according the reviewer’s suggestion.

The mean amplitude (mentioned since the introduction) is a questionable measure because it depends on the spatial relationship between the recording dipole (electrode) and the source dipole (neuron). We absolutely agree with the reviewer and this point was discussed in the first version: “In fact, the amplitudes of extracellular APs decrease rapidly as a function of the distance between the tip of the recording electrode and the soma [35]. However, the relative position between the electrode and soma is not the important factor. In addition, the local field potential depending on the contribution of the global sources of the environment can modify the amplitude [34]”. However, despite this variability, we have observed differences between nuclei.

The recording procedures should all be reported in the Methods section (there is the length of 15-90 s that is reported in the results). Also the descending step on the trajectory is not defined. We have added this sentence in Methods: “Neuronal recordings (Leadpoint®, Minneapolis, USA) of 15-90 s duration were obtained beginning 10 mm above the target and progressing in steps of 0.5 mm. MER (FHC®, Maine, USA) were obtained from both spontaneous and sensory-evoked activity until the inferior border of the thalamus was confirmed by the absence of neuronal activity. Impedance was always above 900 kΩ (1696 ± 80 kΩ, [900, 2900])” (lines 110-114).

The phasic activity properties definition come from an analysis on Parkinson’s patients who are characterized by bursting activity (regular and irregular) in the beta band. It is therefore correct to try other thresholds, but these thresholds should be defined according to the characteristics of the pdf which represent the characteristics of the activity. The global comparison between nuclei is quite unclear. Each nucleus is represented by a set of 6 variables, but these variables are different. The amplitude is a characteristic of the single neuron, the frequency is a characteristic of a neuron in a trace, the density is a characteristic of a trace, the mBI is not defined,  the PI and PR are properties coming from the pdf of a single neuron, but could also come from the pdf of pooled neurons. Therefore, it is impossible to understand whether these are parameters or a set of random variables. From here also comes the definition of the covariance matrix. An exemplary representation of a nucleus should be provided. We have clarified this point at Discussion adding these sentences “We have characterized every nucleus by a point in a 6th dimensional configuration space. This approach is typical of multivariate analysis in which every dimension represents different magnitudes and physical units. Therefore, every point describes the state of the system. It’s quite useful to assess how close or far away are two systems, i.e two points of this space; in other words, we need to describe a metric [36]. (…). As we can observe from equation 7, this metric is dimensionless and therefore, there are not inconsistencies in its use”. We have added a new reference.

The statistical analysis talks about “groups” but this term is not referred to anywhere else. Please explain. Lines 195-200 report the details of the statistical tests performed across “groups”. The expression “Statistical comparisons between groups” is of common use. Nonetheless, we have added this sentence at lines 216-217 “In this context, a group refers to a set of measures of the same variable obtained from the same nucleus, e.g mean amplitude AP, frequency, BI.”

However, no p-values are reported in the Results. We have added the p-values.

The pooled pdfs seem quite similar, and do not discriminate between nuclei. The reviewer is right. We have added these sentences at Discussion to explain this negative result “However, other properties, as mean pdf (figure 2) are not able to discriminate between nuclei. We cannot exclude a bias derived either from the small number of patients or an anesthetic effect. We need a bigger number of patients and compare with pdf obtained from awake patients to explain this fact” (lines 380-383)

As shown in Figure 3, the burst index is not informative because it is defined in such a way that requires the bursts to be present, but in many traces there are no bursts. Regular burts are represented by a pdf with a biphasic distribution (one for the rhythm of the burst and the other one for the rhythm of the spike). This may explain why BI has high variability in the results. Probably this variability (e.g, figure 3d) explain the absence of differentiated pdf between nuclei.

Minor points:

In the introduction, Ce.m.c. and Ce.p.c. are not defined (only Ce is defined). We have defined both acronyms at lines 76-77.

Line 248: interstimulus interval should be inter-spike interval. We have changed the expressions.

Figure 3 refers to the raw trace without spike sorting, correct? Yes

Lines 262-264: revise the sentence. There is a missing verb. Sorry, I cannot find where.

Reviewer 2 Report

This is a very useful subject that could easily mean a big amelioration in the surgical management of patients needing thalamic DBS for epilepsy and other indications. However, I would like to have some points clarified.  Please answer this questions/commentaries in order to finish this report and add some value to this elegant research.

Line 71: more background info is needed here: the anesthesia effect in single unit recording literature review

Line 89: why was proposed the Centro median and no the anterior thalamic? It is part of a clinical protocol? Please explain.

Line 96: BIS?  Please include all the acronyms

Line 97: Please include the antiepileptic medication and when was suspended prior to the surgery (or not)

Line 116: What happened to the vns stimulation during surgery? Turned off? And the battery? Please explain. Also which “implanted programmable stimulator” was used?

Line 121: do you determine the median motor threshold prior to surgery or it is a data from the literature? Please explain  

Line 131: Do you record a period of "silence” after exiting the thalamus? How do you define the thalamus border more specifically?

Line 134: DBS electrode MRI can produce an artefact at contact level. How do you measure your trajectory from the MRI artefact electrode? Please explain more

Line 139: could you define tonic and phasic (bursting?) discharges

Line 214: “assigning 171 mm to the V.c., 99 mm to the Ce and 83 mm to the V.im” : this is the distance of the trajectory inside the nucleus? The normal size of VIM is 4 x 4 x 6 mm.

Line 218: Did you do some spike sorting to identify the single unit recording? Please explain.

Line 233: You say: frequency of action potentials. I would prefer the term firing rate.

Line 248: Interstimulus interval? Did you applied a stimuli during recording? Please explain more clearly this variable.

Line 289: please include the p value for the statistical analysis

Line 302: Are this calculations easily done in a context of very limited time constraint as to be used in surgical theater? How long does it take to do this elegant analysis?

Line 318: Are there any other methods to identify thalamic nucleus during general anesthesia already publish?

Line 329: A neuron's extracellular spike amplitude is seen to be approximately proportional to the sum of the dendritic cross-sectional areas of all dendritic branches connected to the soma. Thus, neurons with many, thick dendrites connected to soma will produce large amplitude spikes. The distance has some influence in the spike width

Line 356: do you have data coming from awake patients? Do they share similarities, what do you find about this in the literature?

Author Response

We are very grateful to the reviewer by its opinions about our work. In addition, we acknowledge the effort made to clarify the manuscript by a detailed reading.

This is a very useful subject that could easily mean a big amelioration in the surgical management of patients needing thalamic DBS for epilepsy and other indications. However, I would like to have some points clarified.  Please answer this questions/commentaries in order to finish this report and add some value to this elegant research.

Line 71: more background info is needed here: the anesthesia effect in single unit recording literature review. We have added a new reference and changed the order of two relevant cites not used at this moment.

Line 89: why was proposed the Centro median and no the anterior thalamic? It is part of a clinical protocol? Please explain. The selection of the target was guided by anatomo-physiological considerations and the reading of papers from Velasco’s school.

Line 96: BIS?  Please include all the acronyms. Done

Line 97: Please include the antiepileptic medication and when was suspended prior to the surgery (or not) We have clarified this point and added a new column in table 1 with AED.

Line 116: What happened to the vns stimulation during surgery? Turned off? And the battery? Please explain. Also which “implanted programmable stimulator” was used?. We have explained these points at lines 91-92 and 125-126

Line 121: do you determine the median motor threshold prior to surgery or it is a data from the literature? Please explain The motor threshold is defined at the beginning of the surgery. Once we check the motor response (e.g, 10 mA), we increase a 50% the stimulus intensity (up to 15 mA). We have indicated that threshold was defined during the surgery (line 132).

Line 131: Do you record a period of "silence” after exiting the thalamus? How do you define the thalamus border more specifically? We have clarified this point adding “The lower border of the thalamus was easily identified by the absence of AP.” (line 129-130).

Line 134: DBS electrode MRI can produce an artefact at contact level. How do you measure your trajectory from the MRI artefact electrode? Please explain more. MRI was used only to identity the lateral and antero-posterior coordinates of the point of electrode. The z coordinate was defined as the lower border of thalamus. Trajectory was reconstructed, as stated in lines 144-145, from the angles used to planning the track.  We have explained in more detail the method in another paper (Pastor, Vega-Zelaya 2019).

Line 139: could you define tonic and phasic (bursting?) discharges. In this line, tonic and phasic refer not to discharges, but to the properties that characterize a discharge and were defined in the literature [24].

Line 214: “assigning 171 mm to the V.c., 99 mm to the Ce and 83 mm to the V.im” : this is the distance of the trajectory inside the nucleus? The normal size of VIM is 4 x 4 x 6 mm. We have clarified this flaw, adding this sentence that illuminate the meaning of the values: “a cumulative trajectory (computed from all the tracks and from all the patients) of” (lines 224-225).

Line 218: Did you do some spike sorting to identify the single unit recording? Please explain. The reviewer is right, and this aspect was absent from Methods. We have added a paragraph in lines 150-155 (2.4 Analysis of discharge properties) explaining this point.

Line 233: You say: frequency of action potentials. I would prefer the term firing rate. Done

Line 248: Interstimulus interval? Did you applied a stimuli during recording? Please explain more clearly this variable. We have changed this expression by inter-spike interval.

Line 289: please include the p value for the statistical analysis. We have added the p-values.

Line 302: Are this calculations easily done in a context of very limited time constraint as to be used in surgical theater? How long does it take to do this elegant analysis?. In our opinion, it would be relatively easy and short-time consuming to implement a program in MATLAB for on-line analysis. For every depth, it takes about 45 s. Considering 10 points, the total amount delayed would be about 8 min. This period is not too much in the context of a DBS surgery.

Line 318: Are there any other methods to identify thalamic nucleus during general anesthesia already publish? We have added a new reference that is the only one similar to our method [31].

Line 329: A neuron's extracellular spike amplitude is seen to be approximately proportional to the sum of the dendritic cross-sectional areas of all dendritic branches connected to the soma. Thus, neurons with many, thick dendrites connected to soma will produce large amplitude spikes. This sentences are taken from references indicated, which used numerical and electrophysiological measurements in animal model.

Line 356: do you have data coming from awake patients? Do they share similarities, what do you find about this in the literature?. At this moment, we are analyzing data from awake patients to compare. Nonetheless, we are at a

Round 2

Reviewer 1 Report

The authors have answered to my previous comments. I only suggest to revise the sentence (lines 282-284) " Therefore, we speculated that perhaps other pairs of times, different from 10/50 ms, could yield better results, taking into account that the mean frequencies obtained were lower than those published for awake patients."

Removing "perhaps" would fix it.

Author Response

We have removed the word "perhaps"